# Is It Possible to Analyze Kidney Functions, Electrolytes and Volemia Using Artificial Intelligence?

**DOI:** 10.3390/diagnostics12123131

**Published:** 2022-12-12

**Authors:** Danijela Tasić, Katarina Đorđević, Slobodanka Galović, Draško Furundžić, Zorica Dimitrijević, Sonja Radenković

**Affiliations:** 1UCC Niš, Clinic of Nephrology, Medical Faculty, University of Niš, 18000 Niš, Serbia; 2Vinča Institute of Nuclear Sciences-National Institute of the Republic of Serbia, University of Belgrade, 11000 Belgrade, Serbia; 3Institute Mihajlo Pupin Belgrade, 11060 Beograd, Serbia

**Keywords:** kidney, heart, electrolytes, congestion, cardio-renal syndrome

## Abstract

Markers used in everyday clinical practice cannot distinguish between the permanent impairment of renal function. Sodium and potassium values and their interdependence are key parameters in addition to volemia for the assessment of cardiorenal balance. The aim of this study was to investigate volemia and electrolyte status from a clinical cardiorenal viewpoint under consideration of renal function utilizing artificial intelligence. In this paper, an analysis of five variables: B-type natriuretic peptide, sodium, potassium, ejection fraction, EPI creatinine-cystatin C, was performed using an algorithm based on the adaptive neuro fuzzy inference system. B-type natriuretic peptide had the greatest influence on the ejection fraction. It has been shown that values of both Na+ and K+ lead to deterioration of the condition and vital endangerment of patients. To identify the risk of occurrence, the model identifies a prognostic biomarker by random regression from the total data set. The predictions obtained from this model can help optimize preventative strategies and intensive monitoring for patients identified as at risk for electrolyte disturbance and hypervolemia. This approach may be superior to the traditional diagnostic approach due to its contribution to more accurate and rapid diagnostic interpretation and better planning of further patient treatment

## 1. Introduction

Renal dysfunction is a common finding in patients with primary and secondary heart disease, and the most common reason for repeated hospitalizations is cardiac decompensation and hypervolemia. It is also known that the therapy used to correct congestion and improve the pumping function of the heart also affects kidney function [1,2]. Therefore, an approach for careful monitoring of renal function and electrolyte levels in addition to assessing volemia status has been included in the guidelines for good clinical practice for the treatment of patients with heart failure. However, the therapy suggested in guideline books often leads to underdosed or underused what due to side effects. The most common side effects of drugs used in the treatment of cardiac decompensation are renal dysfunction and electrolyte disturbance [3,4,5]. The basic parameters for monitoring patients with heart failure are markers of renal function and markers of water–electrolyte balance [6,7]. Sodium (Na+), as an extracellular electrolyte and osmotically active molecule, plays an important role in regulating the water balance. Disorders in serum sodium values are common and are an independent predictor of recurrent hospitalizations due to cardiac decompensation and death after discharge from hospital treatment [8]. Potassium (K+) is an intracellular ion whose role is reflected in the electrical stimulation of muscle and nerve cells. For normal cells to function, a difference between extracellular and intracellular potassium concentration levels must exist. Disorders of serum potassium are common in patients with heart failure. In patients with normal glomerular filtration rate (GFR) values, serum potassium disturbances occur as a part of renin angiotensin aldosterone axis disorder. The disorder is reflected in an imbalance between the sensitivity of tubular cells to aldosterone and the activation of the neurohumoral axis. High mortality has been reported in patients with heart failure who have lower serum potassium values than in those with high serum potassium values [9,10,11]. Assessment of renal function is very important for the assessment of outcomes in patients with primary and secondary heart disease and numerous comorbidities. Deterioration of renal function is associated with frequently repeated hospitalizations and prolonged hospital treatment [12]. In clinical practice, serum creatinine is used daily as a marker to assess the strength of glomerular filtration using various equations. Creatinine is fully filtered in the glomeruli and minimally secreted in the proximal tubules. For this reason, at the present, in practice, creatinine is the best marker of glomerular filtration, with relatively constant plasma concentration. It does not show reliability as a marker of the early stages of acute kidney damage, because its significance depends on the volume state and the intensity of catabolic processes. Glomerular filtration is also assessed using cystatin C in the CKD-EPI creatinine-cystatine C equation (chronic kidney disease epidemiology collaboration). Cystatin C (CyC) is a marker of not only functional, but also structural damage to the kidneys. Cystatin C in patients with essential hypertension can be a marker of subclinical, functional, and structural damage of the heart, as well as a marker of early renal vascular damage. Therefore, cystatin C may be a marker of a subclinical phase of cardiorenal disease [13,14]. Hypervolemia is usually manifested by the appearance of peripheral edema, accumulation of fluid in the abdomen and an increase in intra-abdominal pressure, after an increase in pressure in the right atrium and a decrease in the functional reserve of the glomeruli. The consequent decrease in the functional reserve of the glomerulus occurs due to the activation of the atrial renal reflex during the increase in circulatory volume and the increased filling pressure of the atria. In chronic conditions of hypervolemia, the natriuresis control mechanism regulated by atrial natriuretic peptide and arginine vasoperesin is ineffective and leads to a paradoxical reduction in diuresis. In this case, the reduced intensity of glomerular filtration and diuresis is a consequence of reduced blood flow through the kidney, which occurs due to vasoconstriction of the afferent arteriole after increased sodium absorption in the proximal tubules [15,16]. Natriuretic peptides are biomarkers that have been suggested by guideline books to aid in the noninvasive diagnosis of hypervolemia and heart failure. The determination of the B-type natriuretic peptide (BNP) concentration and its precursor have the greatest significance in the diagnosis of heart failure and are independent predictors of mortality in these patients. Chronic heart failure involves resistance to released NT-proBNP (N-teminal (NT)-pro hormone BNP), as well as deficits in the active from of BNP. NT-proBNP is also elevated in patients who develop acute kidney injury (AKI), due to acute heart failure, since the end-diastolic stretching of cardiomyocytes leads to its production. Elevated levels of NT-proBNP are commonly found in patients with heart failure and reduced glomerular filtration [17,18].

Current research in the field of knowledge-based systems known as synthetic intelligent systems and software algorithms is based on establishing a diagnosis of chronic kidney disease or estimating the time it takes for kidney function to deteriorate using equations for estimating the GFR in pre-dialysis or transplant patients. In some studies, a system was created to monitor the deterioration of kidney function in combination with risk factors that affect the progression of chronic kidney disease. Risk factors for the occurrence of heart diseases were also analyzed and adverse cardiovascular events during percutaneous coronary interventions were evaluated in some studies [19,20,21]. So far, not a single software algorithm has been developed in connection with the prediction of complex problems in patients with combined heart and kidney diseases. Due to the lacking standard of key elements for the diagnosis and prognosis for clinicians to detect cardiorenal syndrome examination, the adaptive neuro fuzzy inference system (ANFIS) proposed in this paper would be a powerful tool to learn the representation of key characteristics for identifying relationships between kidney and heart.

The aim of our paper was to analyze volemia, electrolytes, and renal function in heart failure, using an algorithm based on the ANFIS, an intelligent approach to renal and heart function monitoring [22,23,24,25].

The work is structured in five sections. Section 1 is the introductory part where the basic reasons for the research are presented. Section 2 presents methods in which the research algorithm is presented, and the basic settings of the model are given. Section 3 contains results regarding how data were created and the model was trained. In this section, the statistical inference of the distributions of the input and output flow to the system is presented. Section 4 contains a discussion. The paper ends with conclusions in which directions for future research are given.

## 2. Materials and Methods

### 2.1. Measuring Data

This is a prospective cross-sectional study comparing subjects with associated renal and heart failure or with the existence of a “de novo” or previously diagnosed, clinically manifested cardiovascular disease and with the existence of AKI or the presence of chronic kidney disease at different stages of evolution. The study group included 90 subjects older than 18 years of both sexes with heart and kidney damage; 52 men (57.77%) and 38 women (42.22%).

All patients who had malignant disease of any etiology, acute and chronic inflammatory diseases of other organ systems and clinical manifestations of thyroid disease, were excluded from the study. Blood samples for routine hematological analysis and biochemical analysis after centrifugation for 15 min at 1000 rpm and 5 mL of serum were analyzed by a standard method with commercially available tests. Na+, K+ electrolyte values were measured on a Roche Diagnostics Corporation 9181^®^ Indianapolis, IN, USA analyzer with reference values for Na+ 135–150 mmol/L and for K+ 3.5–5.5 mmol/L. Plasma BNP concentration was determined by enzymatic immunoassay quantitative chemiluminescent microparticle immunoassay (CMIA) technology on an Abbott Laboratories^®^ Germany apparatus. Antiserum-NT-proBNP microparticles were added to the plasma sample, and the reaction was determined as the ratio of the amount of NT-proBNP to the relative light units (RLU). NT-proBNP concentration is expressed in pg/mL. The limit value for NT-proBNP is 300 pg/mL was used as a reference in patients with GFR less than 15 mL/min/1.73 m^2^ calculated using CKD-EPI cystatin C formula. A reference NT-proBNP cutoff value of less than 100 pg/mL was used in patients with GFR if CKD-EPIcystatin C > 90 mL/min/1.73 m^2^. Serum cystatin C (CysC) was determined in plasma using a commercial ELISA (the enzyme-linked immunosorbent assay) kit. Determination of serum cystatin C-based GFR was performed using a reference formula using a calculator [26]. Echocardiographic examinations were performed using a Toshiba Powervision 6000 Tochiba Co^®^ Tokyo Japan device with a multifrequency phase array transducer 2.0–4.5 MHz transthoracic approach in compliance with all recommendations of good clinical practice [27]. This review determined EF% as a functional parameter using the Teicholz formula in M mode or Simpson’s rule in volumetric calculation where normal EF values are greater than 50%, cutoff normal values between 40% and 49%, and low values less than 40% [28].

### 2.2. Neuro-Fuzzy Method

In this paper, we wanted to assess the impact of the occurrence of an imbalance of serum electrolytes (Na+, K+), BNP, ejection fraction (EF), and CKD-EPIcystatin C equations for GFR (glomerular filtration rate) on further monitoring or hospitalization of the patient [29,30,31].

To analyze the given problem, we used the ANFIS-network type, which is supervised learning with fuzzy logic that is similar to Takagi and Sugeno’s approach. The process of learning a neural network with phase logic, shown in Figure 1, represents a complex structural learning of linking input parameters which do not have clearly defined boundaries and their impact with a certain degree of state severity in linking to target values as output parameters [32,33].

### 2.3. Model Description

We used the structure of the ANFIS network, which we based on the connection of input parameters: BNP (pg/mL), Na+ (mmol/L), and K+ (mmol/L) with one output parameter EF (%) or CKD-EPIcystatin C (mL/min/1.73 m^2^) in back propagation (BP), as shown in Figure 2. By normalizing with the min-max method [34,35,36], we adjusted the values of all parameters (input and output) to the range of values of the base parameters [0,1], and thus removed the possibility of dominance of individual data due to approximation and neglect of data values due to different orders of magnitude. Patient data were classified into three groups of ANFIS database data: training data, testing data, and checking data. The structure of the ANFIS network determines the manner and time of training. The network consists of five hidden layers with different numbers of neurons [37,38,39]. The neurons in the layers are related by weighting factors $\omega_{i},i=1,\ldots 27$, which change and adjust during training in the back propagation standard mean square error (MSE). The input layer data (NT-proBNP, Na+ and K+) are adapted due to their range of optimal values by distributing the trapezoidal membership function to the values of the neurons of the next layer of the neural network. The influence of the trapezoidal membership function is such that the value of the input parameter stratifies into three different areas. The NT-proBNP parameter on $\left\{{\mathrm{NT}\text{-}\mathrm{proBNP}}_{1}{,\mathrm{NT}\text{-}\mathrm{proBNP}}_{2}{,\mathrm{NT}\text{-}\mathrm{proBNP}}_{3}\right\}$, Na^+^ on the $\left\{{\mathrm{Na}}_{1}^{+}{,\mathrm{Na}}_{2}^{+}{,\mathrm{Na}}_{3}^{+}\right\}$, and K^+^$\left\{K_{1}^{+}{,K}_{2}^{+}{,K}_{3}^{+}\right\}$, by using reference values described in Section 2.1. The stratified values of individual parameters NT-proBNP, Na^+^ and K^+^ are assigned to the phase of the rule (fuzzy rule) of the form: $$\begin{array}{l} if\left(\mathrm{NT}\text{-}\mathrm{proBNP}\;is\;\mathrm{NT}\text{-}{\mathrm{proBNP}}_{i}\right)\;and\;\left({\mathrm{Na}}^{+}\;\;is\;{\mathrm{Na}}_{j}^{+}\right)\;\;and\;\left(K^{+}\;is\;\;K_{k}^{+}\right)\;then\;\;V_{l}=\;\\ c_{1}\cdot {\mathrm{NT}\text{-}\mathrm{proBNP}}_{i}{+c}_{2}\cdot {\mathrm{Na}}_{j}^{+}{+c}_{3}\cdot K_{k}^{+},\;i,j\;\;ork=\{1,2,3\}and\;\;l=27. \end{array}$$

The third layer normalizes the input value of a single neuron of the third layer with the sum of all values of neurons of the third layer
$${\bar{\omega }}_{l}=\omega_{l}/\sum_{m=1}^{27}\omega_{m=1},l=1,\ldots 27$$

The values normalized in this way are defined by the output membership function (Outputmf), the sum of which determines the value of the output
$$O_{l}={\bar{\omega }}_{l}V_{l},l=1,\ldots 27$$

In the fifth layer, the final value of one output parameter is determined during training as the sum of the values of the fourth layer,
$$\sum_{l=1}^{27}O_{l}=\sum_{l=1}^{27}{\bar{\omega }}_{l}V_{l}$$

The selection of the output parameter is reduced to one and represents either the CKD-EPIcystatin C or EF parameter. The structure of ANFIS requires that the training time of the network is realistic during training in 1000 epochs with a tolerance error for a mean square error (MSE) of 0.0005. This specifically selected structure with the parameters NT-proBNP, Na^+^ and K^+^ leads to the accuracy of the formed network during training, checking on test data (testing data, checking data) where checking is quite consistent and varies in accuracy values of approximately 15% [40,41]. We believe that this is a satisfactory variation in accuracy, although it should be investigated whether the optimization of the proposed algorithm or some other algorithms could achieve a smaller variation in accuracy.

## 3. Results

### 3.1. Implementing the Model

The learning algorithm of ANFIS leads to the formation of a model by connecting the given input and output parameters of the respondents. The ANFIS system formed in this way encourages the use of neural networks in the earlier stages of disruption of individual parameters and indicates the need for faster clinical processing of individual subjects cases. Figure 2 indicates the dependence of one output parameter as a function of two input parameters. The formed three-dimensional surfaces indicate the so-called neuro fuzzy mapping that confirms the following regularities. The area between the green lines indicates the value of K+ clinically stable subjects with certain normalized values of the parameter K+ in the range from 0.20 to 0.60 (area between the green lines), Figure 2a,d,f. Values of Na+ in the range of 0.44 to 1.00 (area between blue lines), Figure 2b,c,e, and NT-proBNP in the range of 0.30 (yellow line) to 0.60 (red line), Figure 2a–c,e EF parameter values below 0.5 (orange horizontal line) indicate patients with a serious adverse event, while EF values above 0.5 indicate patients who are at risk of an adverse event, Figure 2a–c. Values for CKD-EPIcystatin C below 0.58 (pink) indicate renal failure of varying degrees, Figure 2d–f. The dominance of some colors shows that patients with parameters that cause the appearance of yellow colors have heart failure with preserved ejection fraction (HFpEF), while patients who have parameters on horizontal axes that lead to blue colors have heart failure with reduced ejection fraction (HFrEF) and require greater supervision and hospitalization (Figure 2) [42,43,44,45].

### 3.2. Characteristics of the Respondents

In this study, an ANFIS model based on a neural network with fuzzy logic was applied to predict renal function and hydro electrolyte disturbance in patients with heart damage. The usual statistical methods did not find a statistically significant difference in age between healthy subjects who had an average age of 69.55 ± 32.01 years and subjects with heart and kidney damage who had an average age of 70.72 ± 9.26 years (*p* = 0.286). No statistically significant difference was found in the values of electrolyte status parameters shown in Table 1, which includes min, max, mean, and standard deviation (SD) values, in subjects with heart and kidney damage and in healthy subjects. A statistically significant increase in NT-proBNP (*p* < 0.001) and cystatin C (*p* < 0.001) values was found between healthy subjects and subjects with heart and kidney damage (Mann–Whitney U test).

Evaluation metrics are presented in Table 2, which shows the parameters of heart and kidney function in all subjects. A statistically significant difference was found in the values of EF (*p* < 0.001) and CKD-EPIcystatin C equation (*p* < 0.001) by analysis of healthy and diseased subjects (Mann–Whitney U test).

## 4. Discussion

To obtain more accurate results, we used AI to classify the collected data. The central tenet of AI techniques is to computationally automate logical judgment by extracting general rules and patterns from large datasets. The wide range of soft computing techniques are frequently used in system modeling and solving. Cardiorenal syndrome is a complex syndrome characterized by salt and water retention and activation of various neurohumoral mechanisms. Kidney and heart are interconnected by regulatory mechanisms that are important for maintaining homeostasis in the body [46]. Disorder in the function of these mechanisms is an introduction to the vicious circle of causes and consequences, which is characterized by a higher probability of premature death and deterioration of kidney and heart function [47]. Since this outcome is more common in cardiorenal syndrome than if there is isolated heart and kidney damage, it is important to identify high-risk patients as early as possible to apply preventive and therapeutic measures [48].

Type B natriuretic peptide (BNP) is a marker of neurohumoral stimulation whose activity is associated with inhibition of sympathetic nerve activity and the renin angiotensin system axis. NT-proBNP in healthy individuals, even in the case of dietary salt intake, has a protective role for kidney and heart function, while in the early stages of heart and kidney disease it induces natriuresis and diuresis, and in advanced stages of the disease this neurohormone becomes ineffective in regulating hypervolemia. The explanation lies in the fact that at the renal level, NT-proBNP at physiological concentrations acts by increasing the strength of glomerular filtration and directly inhibits the tubuloglomerular feedback response, which first inhibits sodium resorption at the distal tubule, and then at the proximal tubule, reduces intrarenal vascular resistance, but has no effect on the permeability of intrarenal blood vessels [49]. The consequence of the physiological action of the NT-proBNPa molecule is an increase in the volume of excreted urine and an increase in sodium excretion without affecting blood pressure and heart rate [50]. In addition, NT-proBNP plays an important role in the prevention of chronic renal impairment in patients with asymptomatic chronic heart failure due to its effect on intrarenal blood flow. The paradoxical role of NT-proBNP in patients with heart failure by decreased diuresis, natriuresis, and increased vasoconstriction leads to the deterioration of heart and kidney function and general condition of the patient despite a significantly high concentration of the biologically inactive form of circulating BNP [51]. In addition to the fact that the clearance of NT-pro BNP depends on several mechanisms that have not been fully elucidated, it is certain that this protective counter regulatory neurohumoral mechanism is ineffective in patients with heart and kidney damage [52]. The consequences are salt and water retention, hypertension, concentric left ventricular hypertrophy, and heart fibrosis. The endocrine function of the heart could theoretically be improved by delaying the inactivation of cardiac natriuretic peptide hormones and thereby prolonging their beneficial effects [53]. New research at the cellular level, which is related to the low storage capacity for endoproteolytic maturation and processing into biologically active peptides, would be useful in creating a new therapeutic approach compared to the previous one [54].

We have described the typical (and) most common problems in the clinical model of cardiorenal syndrome. Despite the high dynamic nature of progression of cardiorenal syndrome, our model can accurately determine the presence of electrolyte disbalances and hypervolemia. In our study, NT-proBNP was a useful biomarker for assessing the progression of cardiac and renal dysfunction in our subjects with cardiorenal syndrome. The results of our study showed that the overall trend of data verification in the network with NT-proBNP, Na, and K that we formed is approximately 15%, with which subjects can be classified according to the severity of hypervolemia, electrolyte disturbance and renal function [55]. Electrolyte disturbance is a common finding in patients with heart failure and a consequence of the use of diuretics and disorders of neurohumoral activation or a combination of these factors. Hyponatremia is common in patients with acute cardiac decompensation due to dilution and impaired excretion of free water or as a consequence of sodium depletion [56]. Hyperkalemia is often the result of the use of RAAS (the renin-angiotensin-aldosterone system) blockers, mineralocroticode receptor antagonists, or potassium-sparing diuretics. Hypokalemia is also a common finding and is a consequence of magnesium deficiency and the use of Henle loop diuretics [57]. However, in addition to hypokalemia, Henle’s loop diuretics can lead to hypovolemia and deterioration of renal function, which requires a reduction in the administered dose of diuretics, which is the basic and first drug for people with acute cardiac decompensation [58]. There is no standardized method in clinical practice that would detect the degree of decongesting during hospitalization. Therefore, due to the lack of appropriate criteria for defining adequate decongesting, patients require frequent check-ups in an outpatient setting, as in other branches [59]. Assessing the vital risk of patients and the recurrence of decompensation of patients with combined heart and kidney damage involves extensive and repeated diagnosis, with much confusion in terms of determining the causes, consequences, and further treatment planning even by very experienced doctors. This model may be superior to the traditional diagnostic approach due to its contribution to more accurate and rapid diagnostic interpretation and better planning of further patient treatment.

The way in which high values of EPIcistC and EF indicate the risk of adverse events is shown in Figure 2a–f, dependent on the parameters of NT-proBNP, Na+, and K+ patients based on ANFIS results. It has been shown that both low values of Na+ and K+ lead to worsening of the condition and vital endangerment of patients.

To identify the risk of occurrence, the model identifies a prognostic biomarker by random regression from the total data set. This research did not include patients with cardiorenal syndrome who would require additional data preprocessing [59]. They require a different approach and analysis in research that will take radiomics into account, rather than the algorithmic application of a diagnostic methodology.

## 5. Conclusions

Serum potassium disturbances are associated with advanced heart failure and reduced prognosis. The cardiorenal syndrome is used for the estimation of heart failure and kidney disease. There are numerous factors that contribute to the maintenance of disturbed values of potassium in cardiorenal syndrome. Cardiorenal syndrome is definitely independent of many influences, and the balance of serum potassium is more important than sodium in cardiorenal syndrome. In this study, the potassium balance in cardiorenal syndrome was analyzed by the ANFIS. ANFIS is suitable for nonlinear systems with highly redundant data. Although there are encouraging advances around this unsolved clinical problem, further investigation should consider the progressive inclusion of patients with advanced renal impairment to allow a better understanding of the cardiorenal syndrome.

Our work aims to fill a gap by presenting a specific systematized predictive tool for high-risk patients with associated heart and kidney damage. After rigorous validation, this tool will help to predict serious adverse events before they occur and thus improve the treatment outcome of these patients. The predictions obtained from this model can help optimize preventive strategies and intensive monitoring for patients identified as at risk for electrolyte disturbance and hypervolemia.

## Figures and Tables

**Figure 1 diagnostics-12-03131-f001:**
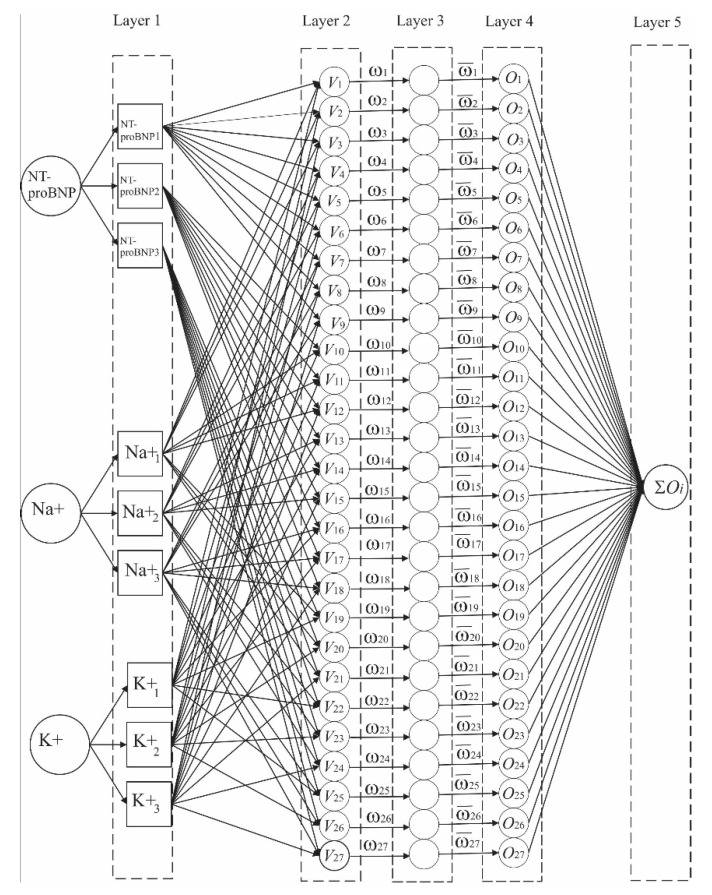
Representation of the ANFIS network used for training on BPU, Na+, K+ parameters with the aim of obtaining EF (%) or CKD-EPIcystatin C as control parameters of cardiac and renal function.

**Figure 2 diagnostics-12-03131-f002:**
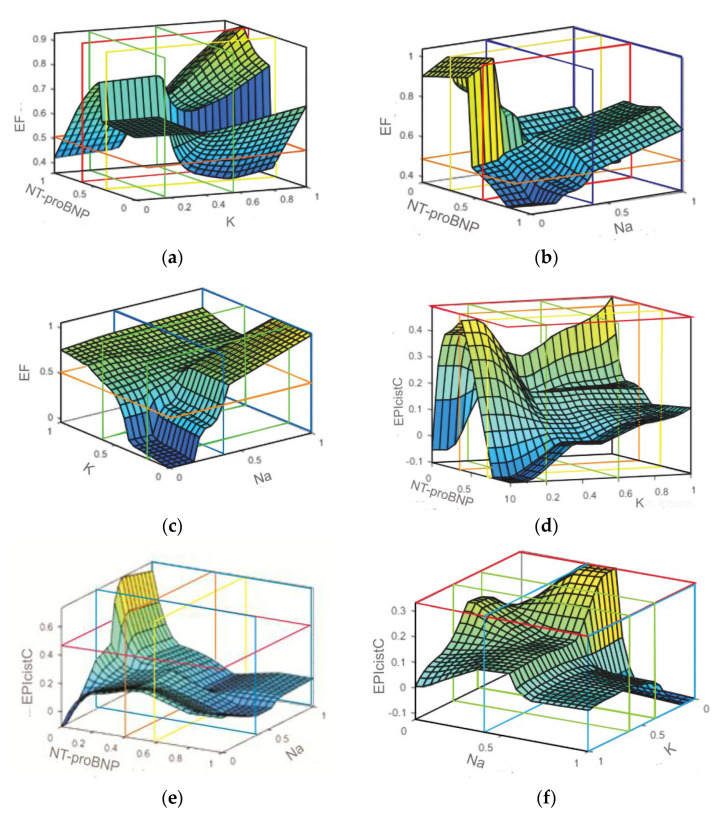
Estimation of ANFIS network of interdependence areas of parameter values: EF values as a functional dependence (**a**) NT-pro BNP and K+, (**b**) BNP and Na+, (**c**) Na+ and K+, and CKD-EPIcystatin C as a functional dependence (**d**) NT-proBNP and K+, (**e**) NT-proBNP and Na+ i (**f**) Na+ and K+.

**Table 1 diagnostics-12-03131-t001:** Demographic and laboratory characteristics of respondents.

Parameters	Na (mmol/L)	K (mmol/L)	NT-pro BNP (pg/mL)	CystatinC (mg/L)	Age (Years)
Min. value	123.00	2.40	10.00	1.73	18.00
Max. value	150.00	7.80	5000.00	0.21	88.00
Mean	137.90	4.84	1275.77	3.33	65.98
SD	4.57	0.97	1533.89	0.825	15.74

**Table 2 diagnostics-12-03131-t002:** Glomerular filtration and functional status of the subjects’ hearts.

Parameters	Min. Value	Max. Value	Mean	SD
Na(mmol/L)	123	150	138	4.12
K(mmol/L)	2.4	7.8	4.85	0.88
NT-pro BNP (pg/mL)	10	5000	1292.10	252.00
EF%	12	75	72.8	15.04
EPI cystatin C(mL/min/1.73 m^2^)	14	146	50.20	37.88

## Data Availability

Documentation and methods used to support this study are available from Danijela Tasic.

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
