# Peer review of "Is It Possible to Analyze Kidney Functions, Electrolytes and Volemia Using Artificial Intelligence?"

_diagnostics, 2022, doi:10.3390/diagnostics12123131_

Round 1
Reviewer 1 Report (Previous Reviewer 2)
Even though the topic is interesting, the authors did not address many of my comments the paper’s novelty is still vague. I can see some standard techniques like ANFIS. Moreover, I asked the authors to add a related works section with a clear discussion of these methods and state their findings and limitations, but they did not. Moreover, English was not revised. There are still many grammatical and punctuation errors. Several more comments are not addressed
The abstract: I cannot see many improvements in the abstract. I cannot find any novelty and you did not highlight the novelty.
Introduction: I can see that the authors added few reference to previous similar studies in the introduction, but without any discussion,The authors are strongly advised to discuss related studies, they should mention their limitations.
English needs to be revised. There are still grammatical and punctuation errors
Again,Could you please highlight the novelty and contribution?
Methods
Table 1 should be discussed in measuring data.
Section 2.2 is entitled :The evaluation metrics used in the study. It does not contain any metrics at all.
I asked for more details regarding the data like How many variables and how many samples? The mean and standard deviation of each variable should be stated.
I asked the authors to justify why didn’t they use deep learning techniques since they are the state of the art, but the authors did not respond
I asked to add a subsection for the evaluation metrics used in this study but with no response.
The figures resolution is still low
Results
The results section should contain more results and analysis.
The authors should have compared ANFIS results with more machine learning techniques.
Author Response
Please see the attached WORD file.

Reviewer 2 Report (Previous Reviewer 1)
Authors answered all comments and suggestions.
Round 2
Reviewer 1 Report (Previous Reviewer 2)
The authors have addressed my comments. I recommend publication of the manuscript
This manuscript is a resubmission of an earlier submission. The following is a list of the peer review reports and author responses from that submission.
Round 1
Reviewer 1 Report
The article takes into consideration the relationship between renal dysfunction and primary or secondary heart disease which is defined as cardiorenal syndrome; specifically, how cardiac damage affects on renal function, which is evaluated analyzing the following parameters: sodium and potassium serum, volemia, BNP and NT-proBNP, ejection fraction and EPI creatinine-cystatin C formula; which are analyzed with an AI system known as ANFIS. The study shows that the serum potassium level is the one that has the greatest impact on cardiorenal syndrome and patient outcome.
My suggestion:
can be added as a keyword: Cardiorenal syndrome.
English language should be improved in both grammar and syntax.
The study included 90 subjects, so it may not be statistically significant.
Lines 272-280 refer to paradoxical role of NT-proBNP in patients with heart failure, but there are no references to recent studies that may have investigated the role of this marker; I recommend to deepen this aspect.
The ANFIS artificial intelligence system leads to the conclusion that serum potassium levels are those that most affect the outcome of the patient with cardiorenal syndrome. I recommend using a different analysis method called radiomics, analyzing the results obtained with the latter and comparing them with those obtained with the ANFIS system. At this regard i can suggest the analysis of this work: htpp://www.ncbi.nlm.nih.gov/pmc/articles/PMC9260602/
Reviewer 2 Report
The paper aims to investigate volemia and electrolytes status from a clinical cardiorenal viewpoint under consideration of renal function using ANFIS s. Generally, the topic is interesting, however, the paper's novelty is vague. I can see some standard techniques. Also, a related works section should be added. English and formatting need to be revised,
The abstract: I cannot see any novelty you are simply using ANFIS neural network to address your question Could you please highlight the novelty in the abstract? The abbreviation ANFIS should be mentioned the first time it appears in the abstract which is not the case in your manuscript. Also, could you please mention the numerical findings of the proposed method?
Introduction: I cannot see any related or previous similar studies in the introduction, The authors are strongly advised to discuss related studies, they should mention their limitations.
There are a lot of typos. Please revise English.
Could you please write the paper organization by the end of the introduction section?
Could you please highlight the novelty and contribution?
Methods
Could you please add samples of the measured data or the mean and standard deviations of the parameters measured?
The paper format needs to be revised. Alignments are not the same.
English needs to be revised.
Could you please add more details regarding the preprocessing methods
My main concern is why the authors did not use deep learning techniques since they are state-of-the-art.
The resolution of the Figures is poor.
How did split that data did you used K-fold cross validation or hold out cross validation?
Also, there should be a subsection for the evaluation metrics used in this study
Discussion
Please mention the limitations of your technique.
The conclusion